# Consistency of Condom Use with Lubricants and Associated Factors Among Men Who Have Sex with Men in Ghana: Evidence from Integrated Bio-Behavioral Surveillance Survey

**DOI:** 10.3390/ijerph22040599

**Published:** 2025-04-11

**Authors:** Ratif Abdulai, Edith Phalane, Kyeremeh Atuahene, Isaiah Doe Kwao, Rita Afriyie, Yegnanew A. Shiferaw, Refilwe Nancy Phaswana-Mafuya

**Affiliations:** 1South African Medical Research Council/University of Johannesburg (SAMRC/UJ) Pan African Center for Epidemics Research (PACER) Extramural Unit, Johannesburg 2006, South Africa; abdullailatif@gmail.com (R.A.); edithp@uj.ac.za (E.P.); 2Department of Research, Monitoring and Evaluation, Ghana AIDS Commission, Accra CT5169, Ghana; skatuahene@yahoo.com (K.A.); isaiah.kwao@ghanaids.gov.gh (I.D.K.); afriyie.rita@ghanaids.gov.gh (R.A.); 3Department of Statistics, Faculty of Sciences, University of Johannesburg, Johannesburg 2006, South Africa; yegnanews@uj.ac.za

**Keywords:** men who have sex with men, consistent condom use with lubricants, HIV, Ghana Men’s Study, condom, lubricant, unprotected sex, anal sex, Ghana

## Abstract

Several studies conducted worldwide have reported on the effectiveness of consistent condom use with lubricants in preventing HIV transmission and acquisition; however, men who have sex with men (MSM) in Ghana continue to be disproportionately affected by the HIV burden. They are stigmatized, discriminated against, and criminalized, leading to social isolation, reduced access to health care, and inadequate targeted interventions. The dissemination of HIV prevention tools such as condoms and lubricants is also mainly focused on the general population, and this approach overlooks the specific needs and vulnerabilities of MSM. This study aimed to determine the prevalence and associated factors of consistent condom use with lubricants among MSM in Ghana. We analyzed cross-sectional data from the Ghana Men’s Study II dataset involving 4095 MSM aged 18 years and above. De-identified data were imported into STATA (College Station, TX, USA, software version 17) for data analysis. Descriptive analysis was performed to describe relevant characteristics of the study population. Multivariable logistic regression analysis was performed for significant variables in bivariate analysis to determine the associated factors of consistent condom use with lubricants. All the statistical analyses were performed at a 95% confidence interval, with significant differences at *p* < 0.05. The prevalence of consistent condom use with lubricants during penetrative anal sex was highest with male partners (44.9%), followed by female partners (40.0%), and all sexual partners (38.9%), respectively. In multivariable logistic regression analysis, having a senior high school education (AOR: 1.76; 95% CI: 0.88–3.12, *p* = 0.039), tertiary education or higher (AOR: 2.24; 95% CI: 0.86–3.23, *p* = 0.041), being an insertive sex partner (AOR: 1.26; 95% CI: 1.02–1.56, *p* = 0.029), being a sex worker (AOR: 1.41; 95% CI: 1.00–1.98, *p* = 0.048), buying sex from other males (AOR: 1.32; 95% CI: 1.03–1.70, *p* = 0.027), being a light drinker (AOR: 0.54; 95% CI: 0.42–0.68, *p* < 0.001), being a moderate drinker (AOR: 0.48; 95% CI: 0.30–0.78, *p* = 0.003), and possessing good HIV knowledge (AOR: 1.79; 95% CI: 1.46–2.20, *p* < 0.001) had higher odds of consistent condom use with lubricants. Being Islamic (AOR: 0.65; 95% CI: 0.49–0.87, *p* = 0.004), having a low income (AOR: 0.57; 95% CI: 0.42–0.77, *p* < 0.001), and easy access (AOR: 0.52; 95% CI: 0.37–0.72, *p* < 0.001) to condoms were positively associated with consistent condom use. This study found a low prevalence of consistent condom use with lubricants among the MSM population in Ghana. The study also found a range of socio-demographic, behavioral, and structural factors associated with consistent condom use with lubricants. This calls for very specific and unique public health interventions, such as developing a predictive model to identify and mitigate barriers to consistent condom use with lubricants.

## 1. Introduction

Recent advancements in the Human Immunodeficiency Virus (HIV) prevention and treatment cascade, such as pre- and post-exposure prophylaxis, have given much hope that the control of HIV as an epidemic in low and middle-income countries is feasible [1,2,3,4,5,6]. The incidence of new HIV infections in most developing countries has seen a significant decline [7]. Accessing HIV care and treatment services has also improved drastically [8,9]. These successes even apply to the least economically developed countries in sub-Saharan Africa (SSA) [2]. However, the HIV epidemiology among specific key population (KP) groups has taken a different dimension in recent times worldwide [10]. Recent evidence indicates that only a little progress has been made in terms of reducing the incidence and prevalence of HIV among men who have sex with men (MSM) from both developing and developed nations [11,12].

Men who have sex with men, particularly those from underdeveloped countries in SSA, have a greater risk of acquiring and transmitting HIV due to stigmatization, discrimination, and criminalization [13,14]. Biologically, the risk of transmitting HIV during unprotected anal sex is about eighteen times higher than in unprotected vaginal sexual intercourse [15,16,17,18,19]. This is further exacerbated by inadequate access to some of the basic HIV prevention materials and care services, such as condoms and lubricants [11,20]. Men who have sex with men often face challenges in accessing these HIV prevention and care services due to the fear of criminalization, discrimination, and denial of services as a result of their sexual orientation [21,22,23]. Studies conducted globally and in some SSA countries have shown that the challenges mentioned above, among others, can negatively impact MSM access to and use of HIV prevention tools such as condoms and lubricants [22,23,24].

Ghana, like many other SSA countries, has a higher HIV prevalence among MSM (18.1%) compared to the general population (1.68%) [25]. Ghana acknowledges MSM in its 2021–2025 National AIDS Strategic Plan [26]; however, drastic measures have not been put in place to meet their HIV prevention and healthcare needs, and this creates a significant barrier to effective healthcare service delivery to the MSM population [27]. National initiatives on HIV prevention are further complicated by socio-cultural norms and stigma by healthcare service providers, and this discourages MSM from seeking necessary health services [27,28]. In a national population-based survey conducted in Ghana, about 90% of Ghanaians expressed that sexual relationships of the same sex are inconsistent with Ghanaian cultural norms and that they undermine the basic social structure of Ghanaian society and are, therefore, unacceptable [28]. In addition to criminalizing same-sex behaviors and creating a socially unfavorable environment, the data regularly show that healthcare providers are also unaware of the unique health needs of MSM in Ghana, resulting in a higher HIV prevalence [25]. The use of condoms with condom-compatible lubricants (CCLs) correctly and consistently has proven to be an effective interventional strategy in reducing HIV transmission and acquisition [29,30,31,32]. Condom-compatible lubricants are those lubricants that enhance the quality and effectiveness of condoms by preventing breakage and making sexual activity very comfortable and more pleasurable [33]. They are mainly water-based lubricants (KY Jelly, Astroglide, and Liquid Silk) and silicon-based lubricants (Wet Platinum, Eros, ID Millennium, etc.) [33]. According to the World Health Organization, silicon and water-based lubricants are safe to use with all kinds of condoms, non-irritating, and do not degrade latex condoms [34]. Oil-based lubricants, on the other hand, are generally not appropriate for safe sex since they cause condoms to weaken, leading to deterioration and, hence, condom breakage, which reduces their effectiveness [35]. Additionally, oil-based lubricants also leave a film on the rectum and the vagina that can trap bacteria and cause their growth, leading to infections [33].

The World Health Organization has reported that using condoms with CCLs consistently and correctly can reduce anal and vaginal transmissions of HIV by over 90% [36]. However, data on the scope of consistent condom use with lubricants among MSM in Ghana are lacking. Only a few studies have looked at condom use among MSM in Ghana [37. Other studies in the sub-region have reported on consistent condom use and their predictors among MSM; however, this has never been reported for the MSM population in Ghana. In a systematic review conducted by Abdulai and colleagues in 2024, only 9 out of 40 studies included in the review reported consistent condom use with lubricants among MSM in SSA [37]. Given the relevance of consistent condom use in HIV prevention, our study went further to determine the prevalence and associated factors of consistent condom use with lubricants among the MSM population in Ghana using a nationwide survey.

Again, several factors associated with consistent condom use among MSM have been reported in different parts of the world [38,39,40,41,42,43]. In a study conducted in India, being exposed to HIV prevention interventions was positively associated with consistent condom use with CCLs [44]. Studies conducted by Ajayi et al. (2019) and Russell et al. (2019) have also documented that discussing HIV/STIs with a sex partner, being aware of the partner’s HIV status, and having had a tertiary or higher education are significant predictors of consistent condom use with CCLs [45,46]. In a study conducted by Abdulai et al. (2024), older age, higher educational level, easy access to condoms and lubricants, risk-reduction counseling, high self-worth, and having the proper knowledge of condoms and lubricant use were reported as significant factors of consistent condom use with CCLs [37]. However, no study has yet been published on the related factors of consistent condom use with lubricants among the MSM population in Ghana.

In this regard, we conducted a retrospective data analysis using integrated bio-behavioral surveillance (IBBS) data from the Ghana Men’s Study II to determine the prevalence of consistent condom use with lubricants and associated factors among the MSM population in Ghana. Findings from this study will serve as evidence for the development of a predictive model to improve the correct and consistent use of condoms with lubricants among MSM and other key population groups in Ghana [47].

## 2. Materials and Methods

### 2.1. Study Design and Study Setting

This study employed a retrospective analysis of the Ghana Men’s Study II data conducted by the Ghana AIDS Commission (GAC) in 2017. The study was conducted across the then-ten (10) regions of the Republic of Ghana to generate comprehensive nationwide data on MSM that can be utilized to address the critical needs of MSM in the Ghanaian population. Ghana is bordered to the north by Burkina Faso, the east by Togo, the west by Ivory Coast, and the south by the Gulf of Guinea. Currently, Ghana has 16 administrative regions, with the Greater Accra Region being the administrative capital. Ghana is known for its natural resources and agriculture, which account for about one-quarter of its gross domestic product.

### 2.2. Study Population

The data for this retrospective analysis involved MSM who participated in the Ghana Men’s Study II. Ghana’s MSM population is about 55,000, with an HIV prevalence of 18.1% as compared to 1.68% among the general populace [48,49]. The Ghana Men Study II included men who were biologically male, aged ≥18, had self-reported consensual sexual intercourse with another male in the past year, and had resided, worked, or socialized in one of the regions where the study was conducted [25]. Transgender women were also considered for the study if they were biologically male and had sex with another male within the last year [25]. In the dataset, “transgender woman” was listed in the options for the question, “What is your sexual orientation/identity?”

### 2.3. Sample Size and Data Collection Process

The Ghana Men’s Study II data were used. The data were collected through IBBSS using a respondent-driven sampling (RDS) technique. Respondent-driven sampling is a popular snowball sampling approach where respondents are drawn from a social network of existing participants in the sample rather than a sampling frame [50]. The Ghana Men’s Study II enrolled a total of 4095 MSM. A sample size of 500 MSM for each region was determined using factors such as projected recruitment time frame, resources, the highly risk nature of such studies, and, most significantly, the ability to measure the most important key indicators using Ghana Men’s Study I estimates with 80% power at a 95% confidence level [25]. The study sample size was also set based on the surveillance objective of tracking significant shifts in the HIV epidemic over time, that is, between rounds of IBBSS. The study treated each location as an independent survey, with the sample size necessary to track any changes at each location [25]. The GAC collected this data through the MSM Comprehensive HIV Prevention Program. The survey questionnaires were administered via a computer, with participants having the alternative to complete them using computer-assisted personal interview software [25]. The data collection process was managed and coordinated by the Bryant Research System to ensure anonymity, privacy, and confidentiality. No personal details or identifiable information that could be traced to the study participants was captured in the data. The data were then captured in the GAC’s electronic database and stored on a secure server with encryption and user access control. Please refer to earlier publications for more detailed information on the sample and data collection process [25].

This retrospective data analysis focuses primarily on reported consistent condom usage with lubricants among MSM with all sexual partners (males and females) and the associated factors of consistent condom use with lubricants among the MSM population in Ghana (Table 1).

### 2.4. Data Analyses

De-identified data were imported into STATA software version 17 (College Station, TX, USA) for data cleaning and processing. Before the statistical analysis, the data were treated for missing information and outliers. “Missing/declined to answer/do not know” cases were excluded from the analysis. Descriptive statistics were performed using univariate analysis to describe relevant characteristics of the study population, such as socio-demographic/socio-economic characteristics, behavioral practices, and condom use behavior. Bivariate analysis was also conducted to determine the relationship between consistent condom use with lubricants among the MSM population and socio-demographic/socio-economic factors and other context-specific factors such as behavioral, clinical, and structural factors. A final multivariable logistic regression analysis was performed to examine the associations of consistent condom use with lubricants. All variables with *p* < 0.05 in the bivariate analysis were used as inputs in a forward stepwise multivariable logistic regression model that was constructed after controlling for confounding variables. The results were presented as adjusted odd ratios (AOR) with their corresponding 95% confidence intervals (CI) and *p*-values. Each independent variable was categorized as seen above for the logistic regression analyses. All the statistical analyses were performed at a 95% confidence interval, with significant differences at α < 0.05; * denotes a *p* < 0.05, ** denotes *p* < 0.01, *** denotes *p* < 0.001.

### 2.5. Ethical Considerations

Formal authorization to access and use the Ghana Men’s Study II data for this study was granted by the Ghana AIDS Commission in writing (Appendix A). This paper forms part of a doctoral study by the first author (Ratif Abdulai, RA) and has received ethics clearance (REC-2742-2024) (Appendix A) from the Research and Ethics Committee of the University of Johannesburg. Additionally, this study also falls under a broader research project at the South African Medical Research Council/University of Johannesburg (SAMRC/UJ) Pan African Centre for Epidemic Research (PACER) Extramural Unit funded project, namely “Harnessing Big Heterogeneous Data to Evaluate the Potential Impact of HIV Responses Among Key Populations in Generalized Epidemic Settings in Sub-Saharan Africa” (REC-1504-2023).

## 3. Results

### 3.1. Socio-Demographic and Socio-Economic Characteristics

Table 2 shows the socio-demographic and socio-economic characteristics of 4095 MSM that participated in the Ghana Men’s Study II. Most MSM were between 18 and 24 years old (2507/3939; 63.65%), of which 37.85% (949/2507) reported consistent condom use with lubricants. Those above 35 years were the least (188/3939; 4.77%), with 35.64% reporting a prevalence of consistent condom use with lubricants. More than half of MSM had at least a secondary level of education (2128/4019; 52.95%). Almost two-fifths of those who had attained a secondary level of education reported using condoms consistently with lubricants (847/2128; 39.80%). A greater number of MSM were single and never married (3810/4047; 94.15%), of which only 38.95% reported consistent condom use with lubricants (1484/3810). For those unemployed (1682/3987; 42.19%), 41.62% (700/1682) reported using condoms consistently with lubricants. Less than one percent reported as being sex workers (28/3987; 0.70%). The majority of MSM were Christians (2812/4035; 69.69%), with a 40.61% prevalence of consistent condom use with lubricants (1142/2812). Few of them were traditionalists (94/4035; 2.33%). Among the traditionalists, 30.85% (29/94) used condoms consistently with lubricants. Over one-third of MSM had no reliable source of income (1430/3860; 36.99%), while only less than a tenth had an income above GHS 1000 per/month (309/3866; 7.99%) (The average USD/GHS exchange rate for 2017 was 4.3996 Ghanaian Cedis per US dollar).

### 3.2. Sexual Orientation and Behavioral Practices Among Men Who Have Sex with Men

Sexual behavior and sexual identity/orientation differ among the MSM population. Out of the total number of 3814 MSM who disclosed their sexual orientation, 43.13% (1645/3814) were gay, 45.67% (1742/3814) were bisexual, and transgender people were the least (29/3814; 0.7%). About one-tenth (10.44%) of them also had other types of sexual orientation. The prevalence of consistent condom use with lubricants among gays, bisexuals, transgender, and other types is 38.84%, 37.54%, 31.03%, and 52.25%, respectively. For sex type, 42.54% of MSM who engaged in insertive sex reported consistent condom use with lubricants. Among MSM who engaged in versatile sex, 33.1% used condoms consistently with lubricants. Less than one-fifth (680/3900; 17.44%) and a quarter (250/992; 25.2%) of those who engaged in transactional sexual intercourse gave money in exchange for sex from males and females in the past six months, respectively. The prevalence of consistency of condom use with lubricants among MSM who engaged in transactional sex with other males and females in the past six months was 33.53% (228/668) and 31.20% (78/250), respectively. The majority of MSM abstained from alcohol, while 2% were heavy drinkers. Among those who abstained from alcohol (2996/4015; 74.6%), 42.2% (1265/2996) of them reported using condoms consistently with lubricants (Table 3).

### 3.3. Condom Use at Last Sex Among Ghanaian Men Who Have Sex with Men

Condom use at the last sex occurrence, with all sexual partners, is presented in Table 4 below. Of the 3411 MSM who reported having a main/regular male partner, about two-thirds (67%) of them reported using condoms at the last sexual intercourse. For those who reported having casual male partners, condom use at the last sex was 66.6%. Among MSM who engaged in transactional sex activities, 67% used condoms when sex was sold to a male in exchange for money and bought sex from a male (63%). Out of the 1195 MSM who reported having main female partners, less than half (46.6%) reported using condoms at the last sexual intercourse with them, 54.5% among casual female partners, and condom use at last sold sex to a female in exchange for money (56.6%). Less than two-thirds of MSM (61.9%) also reported using condoms at the last sex with both males and females.

### 3.4. Consistent Condom with Lubricant Use Among Men Who Have Sex with Men

The primary outcome, “always used a condom with lubricant”, was considered a synonym for consistent condom use with lubricant during sexual intercourse. Consistent condom use with both men and women was reported by 38.9% (*n* = 1559/4008) of MSM, even though 44.9% (*n* = 1795/3996) reported always using condoms with male sex partners during penetrative anal sex (insertive and receptive) only 40.0% (*n* = 1104/2767) reported always using condoms with female sex partners during penetrative anal sex. More than half (55.1%, 2207/4008) of MSM reported using lubricants during anal sexual intercourse (Table 5).

### 3.5. Associated Factors of Consistent Condom Use with Lubricants Among Men Who Have Sex with Men in Ghana

In both bivariate and multivariable regression, having had a senior high school education (AOR: 1.76; 95% CI: 0.88–3.12, *p* = 0.039) and tertiary education (AOR: 2.24; 95% CI: 0.86–3.23, *p* = 0.041) were found to be positively associated with consistent condom use with lubricants. Being Islamic (AOR: 0.65; 95% CI: 0.49–0.87, *p* = 0.004) was positively associated with using condoms with lubricants consistently. Having a low monthly income status (GHS 1–599/month) (AOR: 0.57; 95% CI: 0.42–0.77, *p* < 0.001) and being a sex worker (AOR: 1.41; 95% CI: 1.00–1.98, *p* = 0.048) were the socio-economic factors positively associated with the consistency of condom use with lubricants (Table 6).

With regards to sexual identity/orientation, having an “other type” of sexual identity other than bisexual or transgender was positively associated with consistent use of condoms with lubricants (AOR: 1.83; 95% CI: 1.35–2.49, *p* < 0.001). Being an insertive sex partner (AOR: 1.26; 95% CI: 1.02–1.56, *p* = 0.029) and having bought sex from other males in the past six months (AOR: 1.32; 95% CI: 1.03–1.70, *p* = 0.027) had higher odds for consistent condom use with lubricants.

Similarly, light drinkers (AOR: 0.54; 95% CI: 0.42–0.68, *p* < 0.001) and moderate drinkers (AOR: 0.48; 95% CI: 0.30–0.78, *p* = 0.003) of alcohol were also more likely to use condoms consistently with lubricants in both bivariate and multivariable regression analyses.

Again, having a good knowledge of HIV (AOR: 1.79; 95% CI: 1.46–2.20, *p* < 0.001), as defined by being able to answer the following questions correctly (Is there a cure for HIV?, taking antiretrovirals correctly can improve the lives of people living with HIV, a person can get HIV from witchcraft or other supernatural means, a person can get HIV by receiving an injection with a needle already used by HIV positive person, a person can get HIV by sharing a meal or utensils with someone living with HIV, a person can get HIV through mosquito bites, a health looking person can be HIV positive, people can use condoms consistently to reduce their chances of getting HIV, and people can reduce their chances of getting HIV by having just one sexual partner) was also found to be positively associated with consistent use of condoms with lubricants in both bivariate and multivariable regression analyses (Table 6).

Regarding condom accessibility, easy access (AOR: 0.52; 95% CI: 0.37–0.72, *p* < 0.001) and somewhat easy access (AOR: 0.27; 95% CI: 0.14–0.54, *p* < 0.001) to condoms were also positively associated with using condoms with lubricants consistently in bivariate and multivariable regression. Somewhat easy access was also statistically significantly associated with consistent condom use with lubricants in MSM (Table 6).

## 4. Discussion

### 4.1. Overview

Several studies conducted across the globe have reported on the effectiveness of consistent condom use with lubricants in preventing new HIV transmission and acquisition among MSM [29,30,31,32]. Ghana’s MSM population, like many other countries in SSA, is disproportionately affected by the HIV burden compared to the general population. Stigmatization, discrimination, criminalization, and social isolation make it challenging for most MSM to have access to some of the essential HIV prevention tools, such as condoms and lubricants. Education and dissemination of HIV prevention tools and the health sector response are also tailored mainly towards the general population. Currently, there is no published record on the prevalence of consistent condom use with lubricants among MSM in Ghana. Several factors associated with consistent condom use with lubricants among MSM have also been reported in different parts of the world [38,39,40,41,42,43]; however, this information is also lacking among the MSM population in Ghana. Given the relevance of consistent condom use with lubricants in preventing new HIV transmission and acquisition, this study aimed to determine the prevalence of consistent condom use with lubricants and the associated factors among the MSM population in Ghana.

### 4.2. Main Findings

This study found that less than two-fifths (38.9%) of the MSM participants used condoms with lubricants consistently with all sexual partners. Less than half (44.9%) of the MSM reported consistent condom use with lubricants during penetrative anal sex with other males in the previous six months. Two-fifths (40%) of the MSM participants also reported consistent condom use with lubricants during penetrative anal sex with female partners. Having attained secondary and tertiary education and being Islamic were the socio-demographic factors associated with increased odds for consistent condom use with lubricants with all sexual partners. Consistent condom use with lubricants was also positively associated with being a sex worker and having an income status of 599 cedis/month. Behavioral factors associated with using condoms consistently with lubricants were reported by MSM participants who identified themselves as insertive sex partners and those who bought sex from other males in the previous six months. Light and moderate alcohol drinkers also reported using condoms consistently with lubricants. Similarly, having a good knowledge of HIV also had higher odds of consistent condom use with lubricants. Very easy or easy accessibility to condoms was the only structural factor associated with consistent condom use with lubricants among the MSM participants.

The prevalence of consistent condom use with lubricants (38.9%) reported in this study for MSM with all sexual partners (males and females) is higher than in a survey conducted in Nigeria, which reported just about one-tenth (11%) of MSM using condoms with lubricants consistently with both male and female sexual partners [51]. Other studies conducted in some SSA countries, such as Benin, Nigeria, Malawi, and Swaziland, also reported a low prevalence of consistent condom use with lubricant [52,53,54,55]. The prevalences reported in the above studies are 35%, 21.7%, 29%, and 37% of all those who reported consistent condom use, respectively [52,53,54,55]. Another study conducted in three Southern African countries, Malawi, Namibia, and Botswana, also reported a 38% prevalence of consistent condom use with lubricants [56]. The result of our study for consistent condom use with lubricants (44.9%) with only male partners during penetrative anal sex also corroborates the findings of studies conducted by Zhang and colleagues in three different regions of the Northern part of China and D’anna et al. (2015) [57,58]. These studies reported a 41.1% and 45.2% prevalence of consistent condom use with male partners during anal sex [57,58]. However, the prevalence in our study is lower than those conducted in other parts of China [59,60], which reported a 52% and 56% prevalence of consistent condom use with male partners. Other studies conducted in India and Nigeria also reported a high prevalence of consistent condom use (50% and 53%) during anal sex with male partners than that found in our study [39,51]. A plausible reason for the difference seen in our study compared to those mentioned above may be because, in Ghana, HIV prevention campaigns and distribution of HIV prevention tools are conducted exclusively in the general population, thus making accessibility a challenge for most MSM. It may also be accounted for by the fact that Ghana has punitive measures coupled with stigma and discrimination against homosexual activities, which may have discouraged most MSM from accessing HIV prevention tools in the general population, leading to a low prevalence of consistent condom use with lubricants. The result for consistent condom use with lubricants (40%) during anal sex with female partners is similar to studies conducted in Nigeria by Strömdahl and colleagues and another study conducted in the United States, which reported that 43% and 23.2% of MSM used condoms consistently during anal sex with their female partners respectively [51,53]. Other studies conducted in Japan, Korea, and China also align with our findings [61,62,63].

This study has revealed that having secondary and tertiary levels of education was positively associated with the consistency of condom use with lubricants among the MSM population. This finding is in line with studies conducted by Wang et al. (2021) and Olawore et al. (2021) [59,64]. Men who have sex with men who have at least a senior high school education may have been more aware of the dangers involved in unprotected anal sexual activities and have access to adequate educational resources, which increased their level of self-worth and awareness. Education might have also sharpened their condom-use negotiation skills with their sexual partners, leading to consistent condom use with lubricants.

Our study also found that Islamic people/Muslims were more likely to use condoms with lubricants consistently; however, this does not agree with a study conducted in Ghana among the general populace by Ahinkorah and colleagues in 2020 [65]. The reason for this difference might be that our study focused on the MSM population, whose chance of contracting HIV during unprotected anal sexual intercourse is almost twenty times higher than during unprotected vaginal sex [66,67]. Such a greater risk may have encouraged them to use condoms with lubricants more consistently. The second reason may be attributed to the fact that most Muslim men often have more than one wife and up to four wives in some cases, which implies that having condomless anal sex with other men not only increases their chances of contracting HIV and STIs but their wives as well. This may also have been enough reason for them to use condoms with lubricants more consistently during anal sex with their male counterparts.

For the socio-economic factors, this study found a positive association between consistent condom use with lubricants and MSM who had an income status of 599 cedis/month. This is in consonance with studies conducted in Ghana and Brazil [65,68,69]. Individuals with a moderate income status are also exposed to several HIV prevention campaigns via different avenues (such as newspapers, magazines, and documentaries), which may have helped them to develop strong condom use negotiation skills and decline sex when a condom is not used [70]. Other studies have reported that affluent individuals are more exposed to diverse media opportunities and educational resources and also have access to quality education that keeps them well-informed about recent developments and trends in HIV prevention and control. Such people discuss HIV and STI-related risks more frequently and have a better understanding of the dangers associated with condomless anal sex, have fewer misconceptions, and therefore show more positive attitudes toward condom use [71,72]. Having a moderate income status also elevates people in society, thus increasing their self-worth, and therefore, they choose not to engage in risky sexual behaviors that would jeopardize their health.

Light and moderate drinkers of alcohol were among the behavioral factors positively associated with the consistency of condom use with lubricants among the MSM population in Ghana. This contravenes studies conducted in Japan, China, and the United States [63,73,74,75]. Similar studies conducted by Cai and Lau et al. (2014) and Barrán-Limán et al. (2012) also do not agree with our findings [39,76]; however, it must be stated without any ambiguity that the above studies reported on sex under the influence of alcohol and not just alcohol intake. Sex under the influence of alcohol may have impaired their judgment of safe sexual practices and badly influenced their decision-making, leading to inconsistent condom use [77,78,79], which is not the case in our study. Another reason might be that both light and moderate drinkers of alcohol do not take much alcohol at a time, leaving their judgment of safe sex practices intact or may not take in alcohol before sexual intercourse. The study also revealed a positive association between consistent condom use with lubricants and sex workers, as well as MSM who buy sex from other males. This is supported by studies conducted in the Savannakhet province of the Lao People’s Democratic Republic and Nigeria [80,81]. The study also revealed that insertive anal sex partners used condoms with lubricants consistently. This agrees with the findings of a systematic review conducted by Abdulai et al. (2024) [47]. Among the reasons for the consistent condom use with lubricants may not be limited to reducing anal penetration discomfort and reducing the risk of acquiring or transmitting HIV due to skin tear [82,83].

Consistency of condom use with lubricants was also associated with having a good knowledge of HIV, and this agrees with studies conducted in Cambodia and South Korea [3,61]. Other studies conducted by Mansergh et al. (2006) and Ruan et al. (2008) have also reported a positive association between HIV knowledge and consistent condom use with lubricants [84,85]. Chandra and colleagues also reported in their studies that low HIV knowledge was associated with inconsistent condom use [86]. Men who have sex with men who have good knowledge of HIV are more likely to be well-informed about the dangers associated with condomless anal sexual intercourse, thus making them have a more positive attitude towards consistent condom use with lubricants.

Very easy and easy access to condoms were the structural factors associated with the consistency of condom use with lubricants among the Ghanaian MSM population. This is consistent with studies conducted by Piot et al. (2010), Ramanathan et al. (2013), and Verma et al. (2010) [44,87,88]. Having easy access may have motivated them to use condoms with lubricants consistently to prevent themselves from being infected with HIV or transmitting it to their sexual partners.

### 4.3. Strengths and Limitations

The study contributes to knowledge and addresses a significant gap in understanding sexual health practices among the MSM population in Ghana. This is essential for developing targeted interventions and policies to improve the consistency of condom use with lubricants among the MSM population. The strength of this study relies on the nationwide coverage and reliability of the data. This study is not devoid of limitations, considering that it relies on secondary data. The data used for the analysis relied on self-reporting of information and may be related to biases such as recall and social desirability biases. Because the consistency of condom use with lubricants was self-reported, there is the possibility of under or over-reporting by the MSM participants due to recall bias and among societies that have social norms that frown on condom use. Data for the Ghana Men Study (II) was only collected from men who were biologically male and gave their consent for participation, were aged ≥18, had self-reported consensual sexual intercourse with another male in the past year, and had resided, worked, or socialized in one of the regions where the study was conducted. The “other” represented the sexual identities of men who practiced same-sex behaviors other than gay, bisexual, and transgender women; hence, MSM with other types of sexual orientation or identity apart from what is reported here is not known.

Despite the limitations mentioned above, the Ghana Men’s Study II used rigorous scientific methods employed during the Ghana Men’s Study I to ensure the scientific rigor of the study, comparability, and reliability. The sample size (4095) was calculated to ensure it was large enough to generalize the findings to the entire MSM population in Ghana. It was calculated based on Ghana’s estimated MSM population size. Because the data used for this study were collected in Ghana, its generalizability is only applicable to the Ghanaian MSM population. To determine a suitable sample size for the study, a total sample of 500 MSM was calculated for each of the ten regions involved in the survey. The decision to arrive at the above sample size was made based on the surveillance objective of tracking significant shifts in the HIV epidemic over time, that is, between rounds of IBBSS. The study further treated each location as an independent survey, with the sample size necessary to track any changes at each location. To address selection bias, multiple recruitment techniques were also used to recruit the MSM participants to ensure that different MSM groups were represented in the study, thus broadly enhancing the applicability of the study’s findings to MSM. To further guarantee the reliability and dependability of the study findings, Ghana Men’s Study II employed the same validated instruments as Ghana Men’s Study I.

### 4.4. Practical Implications

The study findings have profound implications for public health interventions, considering that in most SSA countries, condom use is primarily controlled by those who give sex as well as individuals’ demographic and economic factors such as age, marital status, education level attained, socio-cultural norms, and income status. However, receivers of sex may also insist on condom use to protect themselves from contracting HIV and other STIs. Since consistent condom use with lubricants is associated with having at least a secondary level education and partly linked to individual income status, Unique educational interventions such as online education on short courses for MSM are required to provide them with helpful information on HIV prevention, increase self-awareness, and sharpen their condom use negotiation skills. Other interventions should also aim to encourage MSM to participate in capacity-building programs. Condom and lubricant manufacturing companies must be incentivized to produce condom-lubricant pairs to reduce the stress and challenges of accessing both condoms and lubricants. MSM-based organizations must also be included in the national condoms and lubricants distribution process to ensure an adequate supply of condoms and condom-compatible lubricants to its members. Additionally, our study calls for developing a predictive model to improve the consistency of condom use with condom-compatible lubricants. There is also a need for the government of Ghana to prioritize HIV prevention and control in the MSM population by setting aside funds and other resources for HIV prevention and also implementing programs that exclusively target the needs of MSM.

## 5. Conclusions

This study has revealed that over 50% of MSM are not using condoms consistently with lubricants. The prevalence of consistent condom use with lubricants with all sexual partners (males and females), only male partners, and female partners are 38.9%, 44.9%, and 40%, respectively. Demographic and socioeconomic factors such as educational level attained, employment status, religion, and income status influenced consistent condom use with lubricants. Behavioral factors such as light to moderate alcohol intake and having a good knowledge of HIV were also positively associated with the consistency of condom use with lubricants. Easy accessibility to HIV prevention tools, such as condoms and lubricants, is also a significant predictor of consistent condom use with lubricants. To ensure the consistent utilization of condoms with CCLs, condom and lubricant manufacturing companies should be tasked with the responsibility of producing condom-lubricant pairs to reduce the stress of having to look for condoms and lubricants separately. This condom-lubricant pair is my proposed interventional strategy to encourage individuals to use condoms with lubricants consistently. There should also be a differentiated condom and lubricant distribution strategy that includes MSM-based organizations to ensure an adequate supply of condoms and CCLs for its members.

## Figures and Tables

**Table 1 ijerph-22-00599-t001:** Description of study variables.

Research Measures	Indicator
Outcome variable	
Self-reported consistent condom use with lubricants	A Likert-type scale was used with five response categories, namely: Always, Most of the time, Sometimes, Rarely, and Never. Men who have sex with men who reported “always using condoms with lubricants” during sexual intercourse with all male and female partners were considered synonymous with consistent condom use with lubricants. Since a single act of condomless sex might expose one to HIV and other sexually transmitted infections, MSM who reported using condoms with lubricants “most of the time”, “sometimes”, and “rarely” were also classified as inconsistent users. Those who reported “never” to condom use were dropped and excluded from the analysis since our focus is condom users (consistent and inconsistent users). For this analysis, condom use with lubricant variables was recategorized into two; “yes” (consistent users) and “no” (inconsistent users).
Exposure variables	
Types of sexual partners	Types of sexual partners, e.g., regular/main partner (the participant’s devoted partner, such as spouse, lover, or boyfriend); paying partner (person who made payment to the participant in either cash, goods, or services in exchange for sex); selling partner (person who receives payment in the form of money, goods, or services in exchange for sex); casual partner (an unfamiliar individual, a friend, or acquaintance with whom the participant had sex but is not considered a regular or paying partner); all sexual partners (male and female sexual partners)
Socio-demographic factors	Age; Educational level attained (less than primary, primary school, junior high school, secondary, and tertiary/higher); Marital status (single/never married, married/living with a woman, and Widowed/Divorced/Separated); Religion (Christianity, Islam, traditional, other religion, and no religion).
Socio-economic factors	Employment status (unemployed, employed in the informal sector, formal sector, and sex worker); Income status/month (no reliable source of income, low income, middle income, and high income).
Behavioral factors	Sexual identity/orientation (gay, bisexual, transgender, and other); Sex type (insertive anal sex, receptive anal sex, and versatile); Transactional sex (bought sex from a male in the past six months, bought sex from a female in the past six months, sold sex to a male in the past six months, sold sex to a male in the past six months, and alcohol intake); Alcohol intake (abstainers, light drinkers, moderate drinkers, and heavy drinkers); Good HIV knowledge (yes or no);
Clinical factors	Syphilis status (positive or negative); HIV test results (positive or negative); Hepatitis B status (positive or negative); Knowing one’s HIV status (yes or no).
Structural factors	HIV counseling (yes or no); Condom accessibility (very easy, easy, and somewhat easy); Condom affordability (very affordable, somewhat affordable, and expensive).

**Table 2 ijerph-22-00599-t002:** Demographics and Socio-economic Characteristics of the Study Participants.

Variables	Consistent Condom Use with Lubricants		Total % (*n*)
	Yes *n* (%)	No *n* (%)	
Age category			
18–24	949 (37.85)	1558 (62.15)	2507 (63.65)
25–34	488 (39.23)	756 (60.77)	1244 (31.58)
35+	67 (35.64)	121 (64.36)	188 (4.77)
Total	1504 (38.18)	2435 (61.82)	3939 (100.00)
Educational level attained			
Less than primary	49 (28.32)	124 (71.68)	173 (4.30)
Primary school	52 (37.41)	87 (62.59)	139 (3.46)
Junior High school	397 (36.22)	699 (63.78)	1096 (27.27)
Secondary school	847 (39.80)	1281 (60.20)	2128 (52.95)
Tertiary or higher	214 (44.31)	269 (55.69)	483 (12.02)
Total	1559 (38.79)	2460 (61.21)	4019 (100.00)
Marital status			
Single/Never Married	1484 (38.95)	2326 (61.05)	3810 (94.15)
Married/living with a woman	59 (32.96)	120 (67.04)	179 (4.42)
Widowed/Divorced/Separated	22 (37.93)	36 (62.07)	58 (1.43)
Total	1565 (38.67)	2482 (61.33)	4047 (100.00)
Employment			
Unemployed	700 (41.62)	982 (58.38)	1682 (42.19)
Informal	349 (39.57)	533 (60.43)	882 (22.12)
Formal	288 (34.45)	548 (65.55)	836 (20.97)
Sex worker	14 (50.0)	14 (50.0)	28 (0.70)
Other	202 (36.14)	357 (63.86)	559 (14.02)
Total	1553 (38.95)	2434 (61.05)	3987 (100.00)
Religion			
Christianity	1142 (40.61)	1670 (59.39)	2812 (69.69)
Islamic	139 (26.99)	376 (73.01)	515 (12.76)
Traditional	29 (30.85)	65 (69.15)	94 (2.33)
Other	205 (43.80)	263 (56.20)	468 (11.60)
No religion	49 (33.56)	97 (66.44)	146 (3.62)
Total	1564 (38.76)	2471 (61.24)	4035 (100.00)
Income status (GHS)			
No Income	643 (44.97)	787 (55.03)	1430 (36.99)
Low Income (1–599 cedis/month)	660 (35.95)	1176 (64.05)	1836 (47.49)
Middle Income (600–999 cedis/month)	106 (36.43)	185 (63.57)	291 (7.53)
High come (≥1000 cedis/month)	115 (37.22)	194 (62.78)	309 (7.99)
Total	1524 (39.42)	2342 (60.58)	3866 (100.00)

The average USD/GHS exchange rate for 2017 was 4.3996 Ghanaian Cedis per US dollar.

**Table 3 ijerph-22-00599-t003:** Sexual orientation and behavioral practices among men who have sex with men.

Variables	Consistent Condom Use with Lubricants		Total % (*n*)
	Yes *n* (%)	No *n* (%)	
Sexual identity/orientation			
Gay	639 (38.84)	1006 (61.16)	1645 (43.13)
Bisexual	654 (37.54)	1088 (62.46)	1742 (45.67)
Transgender	9 (31.03)	20 (68.97)	29 (0.76)
Other	200 (50.25)	198 (49.75)	398 (10.44)
Total	1502 (39.38)	2312 (60.62)	3814 (100.00)
Type of anal sexual intercourse			
Insertive anal sex	661 (42.54)	893 (57.46)	1554 (44.41)
Receptive anal sex	350 (40.46)	515 (59.54)	865 (24.72)
Versatile sex	357 (33.06)	723 (66.94)	1080 (30.87)
Total	1368 (39.10)	2131 (60.90)	3499 (100.00)
Bought sex from a male in the past six months			
Yes	228 (33.53)	452 (66.47)	680 (17.44)
No	1283 (39.84)	1937 (60.16)	3220 (82.56)
Total	1511 (38.74)	2389 (61.26)	3900 (100.00)
Bought sex from a female in the past six months			
Yes	78 (31.20)	172 (68.80)	250 (25.20)
No	228 (30.73)	514 (69.27)	742 (74.80)
Total	306 (30.85)	686 (69.15)	992 (100.00)
Sold sex to a male in the past six months			
Yes	433 (37.49)	722 (62.51)	1155 (29.86)
No	1059 (39.03)	1654 (60.97)	2713 (70.14)
Total	1492 (38.57)	2376 (61.43)	3868 (100.00)
Sold sex to a female in the past six months			
Yes	42 (28.57)	105 (71.43)	147 (15.03)
No	257 (30.93)	574 (69.07)	831 (84.97)
Total	299 (30.57)	679 (69.43)	978 (10.00)
Alcohol intake			
Abstainers	1265 (42.22)	1731 (57.78)	2996 (74.62)
Light drinker	229 (28.99)	561 (71.01)	790 (19.68)
Moderate drinker	36 (24.00)	114 (76.00)	150 (3.73)
Heavy drinker	22 (27.85)	57 (72.15)	79 (1.97)
Total	1552 (38.66)	2463 (61.34)	4015 (100.00)

**Table 4 ijerph-22-00599-t004:** Condom use behavior at last sex with different sexual partners among Ghanaian men who have sex with men.

Variables	Number (*n*)	Percentage (%)
Condom use at last sex with main/regular male partner ^a^		
No main/regular male partner	550	13.9
Yes	2288	67.1
No	1123	32.9
Sub-total	3411	100.0
Condom use at last sex with a casual male partner ^b^		
No casual male partner	1588	40.2
Yes	1574	66.6
No	789	33.4
Sub-total	2363	100.0
Condom use at last sold sex to a male in exchange for money ^c^		
Have not sold sex to a male in exchange for money	2325	58.8
Yes	1091	67.0
No	538	33.0
Sub-total	1629	100.0
Condom use at last bought sex with a male ^d^		
Do not have a male from whom I bought sex in exchange for money	2856	72.03
Yes	699	63.0
No	410	37.0
Sub-total	1109	100.0
Condom use at last sex with a male from whom you received gifts or goods ^e^		
Do not have a man I received gifts or goods in exchange for sex	2574	64.8
Yes	948	67.9
No	449	32.1
Sub-total	1397	100.0
Condom use at last sex with a male to whom you gave gifts or goods for sex ^f^		
Do not have a man I gave gifts or goods in exchange for sex	2946	74.3
Yes	648	63.7
No	370	36.3
Sub-total	1018	100.0
Condom use at last sex with a main female partner ^g^		
Do not have a female partner	1743	43.9
Do not have a main female partner	315	7.9
Yes	893	46.6
No	1022	53.4
Sub-total	1915	100.0
Condom use at last sex with a casual female partner ^h^		
Do not have a female partner	1743	43.7
Do not have a casual female partner	1259	31.6
Yes	537	54.5
No	448	45.5
Sub-total	985	100.0
Condom use at last sold sex to a female for money ^i^		
Have not sold sex to a female in exchange for money	3344	83.7
Yes	368	56.6
No	282	43.4
Sub-total	650	100.0
Condom use at last sex with a man or a woman		
Yes	2409	61.9
No	1482	38.1
Total	3891	100.0

Sub-total is the sum of “Yes and No” responses; ^a^ an analysis based on a sub-sample of 3411 MSM who had sex with a main/regular male partner; ^b^ an analysis based on a sub-sample of 2363 MSM who had sex with a casual male partner; ^c^ an analysis based on a sub-sample of 1629 MSM who sold sex to a male; ^d^ an analysis based on a sub-sample of 1109 MSM who bought sex from a male; ^e^ an analysis based on a sub-sample of 1397 MSM who had sex with a male from whom they received gifts or goods; ^f^ an analysis based on a sub-sample of 1018 MSM who had sex with a male to whom they gave gifts or goods; ^g^ an analysis based on a sub-sample of 1915 MSM who had sex with a main female partner; ^h^ an analysis based on a sub-sample of 985 MSM who had sex with a casual female partner; ^i^ an analysis based on a sub-sample of 650 MSM who sold sex to a female and ‘n’ is the frequency.

**Table 5 ijerph-22-00599-t005:** Consistent condom use with lubricants among men who have sex with men.

Variables	Number (*n*)	Percentage (%)
How often do you use condoms with lubricants when you have sex with a man or a woman?		
Always	1559	38.9
Most of the time	824	20.6
Sometimes	985	24.6
Rarely	286	7.1
Never	354	8.8
Total	4008	100.0
How often do you use condoms with lubricants when you have penetrative anal sex with other males?		
Always	1795	44.9
Most of the time	505	12.6
Sometimes	870	21.8
Rarely	244	6.1
Never	582	14.6
Total	3996	100.0
How often do you use condoms with lubricants when you have penetrative anal sex with females?		
Always	1104	40.0
Most of the time	271	9.8
Sometimes	621	22.4
Rarely	236	8.5
Never	535	19.3
Total	2767	100.0
How often do you use lubricants for anal sex?		
Always	2207	55.1
Most of the time	507	12.6
Sometimes	636	15.9
Rarely	105	2.6
Never	553	13.8
Total	4008	100.0

**Table 6 ijerph-22-00599-t006:** Factors associated with consistent condom use with lubricants among men who have sex with men in Ghana.

Variables	% (*n*)	Bivariate Analysis		Multivariable Analysis	
		Unadjusted Odds Ratio (95% CI)	*p*-Value	Adjusted Odds Ratio (95% CI)	*p*-Value
Age category					
18–24	62.96 (1406)	REF			
25–34	33.32 (744)	1.09 (0.91–1.31)	0.333		
35+	3.72 (83)	1.07 (0.68–1.69)	0.774		
Educational level					
Less than primary	2.28 (51)	REF		REF	
Primary school	3.00 (67)	1.55 (0.92–2.57)	0.093	1.16 (0.52–2.61)	0.717
Junior High school	26.11(583)	1.47 (1.00–2.15)	0.052	1.58 (0.83–3.00)	0.164
Senior High school	55.44 (1238)	1.66 (1.15–2.42)	0.006 *	1.76 (0.88–3.12)	0.039 *
Tertiary or higher	13.17 (294)	1.99 (1.33–3.00)	0.012 *	2.24 (0.86–3.23)	0.041 *
Marital status					
Married/living with a woman	4.75 (106)	REF			
Single/Never Married	94.04 (2100)	1.42 (0.93–2.17)	0.103		
Widowed/Divorced/Separated	1.21 (27)	1.59 (0.66–3.80)	0.298		
Employment					
Unemployed	39.23 (876)	REF		REF	
Employed	45.77 (1022)	0.75 (0.62–0.90)	0.002 **	1.17 (0.85–1.59)	0.336
Sex worker	15.00 (335)	0.83 (0.64–1.07)	0.150	1.41 (1.00–1.98)	0.048 *
Religion					
Christianity	73.13 (1633)	REF		REF	
Islamic	12.94 (289)	0.64 (0.48–0.84)	0.001 **	0.65 (0.49–0.87)	0.004 **
Traditional	1.75 (39)	0.69 (0.35–1.37)	0.289	0.78 (0.38–1.60)	0.496
Other	9.40 (210)	1.19 (0.89–1.59)	0.249	1.14 (0.83–1.55)	0.418
No religion	2.78 (62)	0.54 (0.30–0.96)	0.036 *	0.66 (0.36–1.21)	0.181
Income status (GHS)					
No Income	32.38 (723)	REF		REF	
Low Income (1–599 cedis/month)	50.47 (1127)	0.58 (0.48–0.70)	<0.001 ***	0.57 (0.42–0.77)	<0.001 ***
Middle Income (600–999 cedis/month)	7.61 (170)	0.73 (0.52–1.02)	0.069	0.65 (0.42–1.02)	0.062
High come (≥1000 cedis/month)	9.54 (213)	0.78 (0.57–1.07)	0.126	0.75 (0.49–1.16)	0.195
Sexual identity/orientation					
Gay	41.51 (927)	REF		REF	
Bisexual	47.11 (1052)	0.85 (0.71–1.02)	0.083	0.89 (0.73–1.09)	0.275
Transgender	0.31 (7)	4.08 (0.79–21.16)	0.094	5.40 (1.00–29.27)	0.050
Other	11.06 (247)	1.70 (1.28–2.26)	<0.001 ***	1.83 (1.35–2.49)	<0.001 ***
Type of anal sexual intercourse					
Versatile sex	35.29 (788)	REF		REF	
Receptive sex	22.03 (492)	1.19 (0.94–1.51)	0.141	1.11 (0.86–1.44)	0.403
Insertive sex	42.68 (953)	1.37 (1.12–1.66)	0.002 **	1.26 (1.02–1.56)	0.029 *
Number of receptive partners					
One	73.00 (1630)	REF			
Two or more	27.00 (603)	0.85 (0.70–1.03)	0.102		
Number of insertive partners					
One	62.52 (1396)	REF			
Two or more	37.48 (837)	1.02 (0.86–1.22)	0.786		
Bought sex from a male in the past six months					
Yes	19.03 (425)	REF		REF	
No	80.97 (1808)	1.54 (1.22–1.93)	<0.001 ***	1.32 (1.03–1.70)	0.027 *
Sold sex to a male in the past six months					
Yes	31.30 (699)	REF			
No	68.70 (1534)	1.01 (0.84–1.21)	0.934		
Alcohol intake					
Abstainers	71.56 (1598)	REF		REF	
Light drinkers	21.72 (485)	0.50 (0.40–0.63)	<0.001 ***	0.54 (0.42–0.68)	<0.001 ***
Moderate drinkers	4.70 (105)	0.43 (0.27–0.68)	<0.001 ***	0.48 (0.30–0.78)	0.003 **
Heavy drinkers	2.02 (45)	0.62 (0.33–1.17)	0.141	0.69 (0.35–1.35)	0.278
Good knowledge of HIV					
No	32.69 (730)	REF		REF	
Yes	67.31 (1503)	1.80 (1.49–2.18)	<0.001 ***	1.79 (1.46–2.20)	<0.001 ***
Syphilis status					
Positive	0.99 (22)	REF			
Negative	99.01 (2211)	0.50 (0.22–1.17)	0.109		
Hepatitis B (HBsAg) status					
Positive	7.57 (169)	REF			
Negative	92.43 (2064)	0.82 (0.60–1.13)	0.235		
Knowing one’s HIV status					
Yes	17.64 (394)	REF			
No	82.36 (1839)	1.00 (0.80–1.25)	0.976		
Condom accessibility					
Very easy	84.24 (1881)	REF		REF	
Easy	12.09 (270)	0.53 (0.39–0.70)	<0.001 ***	0.52 (0.37–0.72)	<0.001 ***
Somewhat easy	3.67 (82)	0.23 (0.12–0.43)	<0.001 ***	0.27 (0.14–0.54)	<0.001 ***
Affordability of condoms					
Very affordable	81.42 (1181)	REF		REF	
Somewhat affordable	13.12 (293)	0.75 (0.58–0.98)	0.033 *	1.19 (0.87–1.63)	0.265
Expensive	5.46 (122)	0.76 (0.52–1.13)	0.177	1.20 (0.78–1.84)	0.416

Abstainer means do not consume alcohol; light drinker means 1–3 drinks/week; moderate drinker means 4–5 drinks/week; and heavy drinker means 6–12 drinks/week. * Means *p* < 0.05; ** means *p* < 0.01; ***—*p* < 0.001; REF—Reference category, ‘n’—frequency. The average USD/GHS exchange rate for 2017 was 4.3996 Ghanaian Cedis per US dollar. CI—Confidence interval.

## Data Availability

Data can only be acquired from the Ghana AIDS Commission at info@ghanaids.gov.gh upon reasonable request.

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
