# Peer review of "Consistency of Condom Use with Lubricants and Associated Factors Among Men Who Have Sex with Men in Ghana: Evidence from Integrated Bio-Behavioral Surveillance Survey"

_ijerph, 2025, doi:10.3390/ijerph22040599_

Round 1
Reviewer 1 Report
Comments and Suggestions for Authors
The article addresses an important public health issue by focusing on HIV prevention among men who have sex with men (MSM) in Ghana, emphasizing the use of condoms and lubricants. It takes a bold and unconventional approach to fill a critical gap in the literature. HIV prevention strategies targeting the MSM community are of paramount importance for both local and international public health. I commend the researchers for their efforts but have some concerns. Below are my recommendations for improvement:
-The abstract is overly long and should be shortened for clarity and conciseness.
-The introduction should include more information on the social, cultural, and legal challenges faced by the MSM population in Ghana. This would provide better context and enhance the reader's understanding of the study's significance.
-A dedicated paragraph is needed to clearly articulate how this study differs from previous research and addresses a gap in the existing literature.
-The explanation of data collection and analysis is too general and lacks specificity. Details about how "missing data" and "outliers" were managed should be explicitly included.
-The discussion section is limited in its comparison with the existing literature and does not sufficiently address the practical implications of the findings. More emphasis should be placed on how the results inform public health policies, HIV prevention strategies, and actionable recommendations for the MSM community.
-The referenced literature should be updated to include studies conducted in the past five years, particularly those that are regionally relevant or address similar issues in other low- and middle-income countries.
-The study’s focus on the MSM population in Ghana limits its generalizability. This limitation should be explicitly discussed, along with its implications for broader contexts.
-The conclusion is brief and lacks impact. It should clearly outline the study’s contributions to the broader literature and propose more specific, actionable recommendations.
Reviewer 2 Report
Comments and Suggestions for Authors
Detailed suggestions are included in the peer-review report.

Minor suggestions are included in the peer-review report.
Reviewer 3 Report
Comments and Suggestions for Authors
The study addresses the problematic of consistent condom use with condom-appropriate lubricants among the male population in Ghana who has acknowledged of having had sex with another male. The reason for the necessity of such a study is the very high prevalence of HIV infections among the male population which is engaging in same-sex intercourse.
The paper is highly relevant due to some results diverging from existing studies with the same focus in other geographical areas. It needs some proof-reading and editing. Please see attached file for some suggestions of improvement.

Round 2
Reviewer 1 Report
Comments and Suggestions for Authors
Dear Editor,
I have reviewed the revised version of the manuscript titled "Consistency of Condom Use with Lubricants and Associated Factors Among Men Who Have Sex with Men in Ghana", for which I previously requested a major revision. The authors have carefully addressed my comments and have diligently implemented the suggested revisions. The changes made have significantly improved the scientific quality and clarity of the study.
These revisions have improved the manuscript’s overall coherence and academic contribution. I find the changes satisfactory and recommend the manuscript for publication in its current form. Best regards.